# SHARCS: Efficient Transformers through Routing with Dynamic Width Sub-networks

**Mohammadreza Salehi**[†] **Sachin Mehta**[†] **Aditya Kusupati**[†]
**Ali Farhadi**[†◇] **Hannaneh Hajishirzi**[†◇]

[†]University of Washington   [◇]Allen Institute for Artificial Intelligence
{mrsalehi,sacmehta,kusupati,ali,hannaneh}@cs.washington.edu

## Abstract

We introduce SHARCS for adaptive inference that takes into account the hardness of input samples. SHARCS can train a router on any transformer network, enabling the model to direct different samples to sub-networks with varying widths. Our experiments demonstrate that: (1) SHARCS outperforms or complements existing per-sample adaptive inference methods across various classification tasks in terms of accuracy vs. FLOPs; (2) SHARCS generalizes across different architectures and can be even applied to compressed and efficient transformer encoders to further improve their efficiency; (3) SHARCS can provide a $2\times$ inference speed up at an insignificant drop in accuracy.

## 1 Introduction

Web-scale pretrained models, including Large Language Models (LLMs), are widely used in various applications (Devlin et al., 2018; Liu et al., 2019a; Brown et al., 2020). However, their computational resource requirements can be problematic, especially in environments with limited resources. To address this issue, more efficient methods are needed, particularly those that can run on-device with efficient inference (Sanh et al., 2019).

Several methods (e.g., knowledge distillation (Hinton et al., 2015), pruning (Lagunas et al., 2021; Xia et al., 2022), and quantization (Shen et al., 2019)) have been proposed to improve the inference efficiency of transformer-based models. While these methods are promising, one drawback is that the resulting model is static. This raises concerns about whether the model is too complex for simple samples and not complex enough for more challenging ones. To tackle this problem, previous work have investigated sample-adaptive inference methods that use varying amount of compute to process different input samples (Kaya and Dumitras, 2018). Two predominant approaches exist in the field of sample adaptive inference: early-exiting

and token dropping. The former incorporates internal classifiers into intermediate layers of the model. Various techniques have been explored for early exiting, including using confidence scores or entropy of internal classifier predictions (Liu et al., 2020; Xin et al., 2020), using a module that predicts whether a layer should exit early (Xin et al., 2021), or implementing a patience-based method to adjust these internal predictions (Zhou et al., 2020). The other category of sample-adaptive inference methods, token dropping, enhances efficiency by progressively decreasing sequence length as the forward pass proceeds layer by layer (Goyal et al., 2020; Guan et al., 2022).

In this paper, we propose **S**ample **H**ardness **A**ware **R**outing based on **C**onfidence **S**cores (SHARCS[1]), which is a novel category within the efficient sample adaptive inference domain. Our approach introduces *training a light-weight router*. The router dynamically assigns input samples, based on their hardness (Ethayarajh et al., 2021), to one of the *sub-networks with varying widths*. Due to the lack of ground truth notion of hardness, we estimate sample hardness heuristically based on the network's prediction history during training for each sample. These estimates are then used as labels to train the router.

We make the following contributions:

1. SHARCS introduces a router that predicts the hardness of a sample and can be trained on *any* transformer network. It enables dynamic input processing using sub-networks of varying widths, determined by the predicted hardness.

2. SHARCS delivers substantial efficiency improvements in terms of accuracy vs. FLOPs trade-off across different datasets & transformer encoders. For example, on QQP (Wang et al., 2017), it reduces the FLOPs of RoBERTa_base by $2.75\times$ with only $1\%$ accuracy drop. Compared to other

---

[1]pronounced sharks.

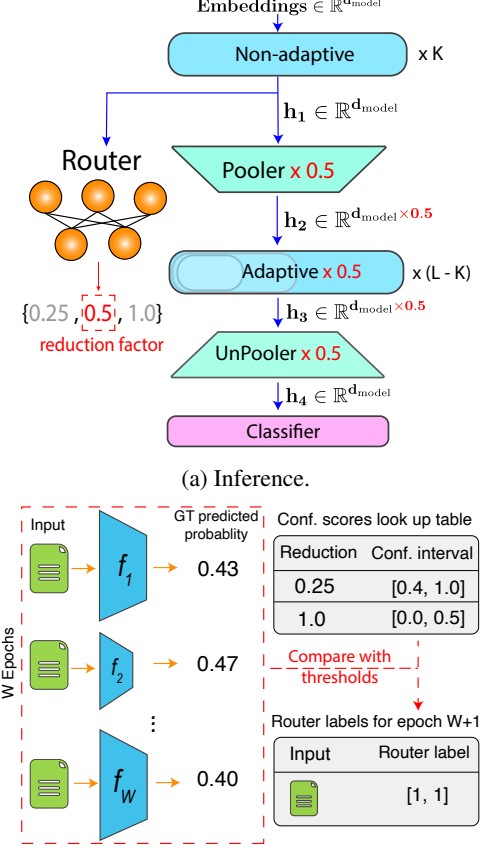

(a) Inference.

(b) Generating hardness labels to train the router.

Figure 1: **(a)** At inference the router selects the reduction factor. Red parts denote the changes enforced by the router. **(b)** Training the router. The confidence scores of the last $W$ epochs for each input sample is recorded. Then they are compared with confidence thresholds and labels for training the router are assigned accordingly.

sample adaptive inference techniques, SHARCS either outperforms them or can be paired with them to achieve even greater efficiency.

3. The gains from SHARCS can be realized in real-world deployment with significant latency reduction in CPU-based inference. On QQP, SHARCS can speed up BERT$_{\text{base}}$ more than $2\times$ with less than 1% drop in accuracy.

## 2  Method

We introduce SHARCS for adaptive inference that takes into account the hardness of input samples. Our approach has three main steps: (1) Obtaining labels that represent sample hardness (§2.1); (2) Training a router that can predict sample hardness (§2.2); (3) Adjusting the network's inference capacity according to the predicted hardness (§2.3).

### 2.1  Estimating Hardness of a Sample

Our objective is to learn a sample hardness aware router, enabling us to dynamically direct the input sample to one of the sub-networks with varying capacities (including the main network) for efficient inference. As there are no ground-truth labels that represent hardness, we leverage network's prediction history during training. We assume that there are $M$ possible hardness levels for a sample, with level 1 being the easiest and $M$ being the hardest. Our goal is to assign a hardness label $\hat{y} \in \{0, 1\}^M$ to samples in the training set and train the router with them. Toward this end, we employ a heuristic procedure which is illustrated in Figure 1b: If the model predicts the ground truth class for a training sample with a probability within a confidence interval $[T^i_{low}, T^i_{high}]$, then the $i$th entry in the label $\hat{y}$ would be 1; otherwise it would be zero. Here, $i$ ranges from 1 to $M$, and $T^i$'s are hyperparameters associated with hardness level $i$.

Because of the stochastic nature of training, it is possible that the samples denoted as easy earlier will potentially be denoted as hard after that, posing instabilities while training the router. To mitigate such randomness, the $i$th entry is 1 only if the predicted ground truth probability is within the interval for a moving window of last $W$ epochs. The assigned hardness labels will be used as labels to train the router in the next epoch. We do not train the router during the first $W$ epochs and just train the network and record the network's predictions. Please see appendix E.0.5 for more details on the role of window size.

### 2.2  Training Sample Hardness Aware Router

We split the main transformer network with $L$ layers into non-adaptive and adaptive parts, and incorporate the router between these networks (see Figure 1a). The non-adaptive network is comprised of the first $0 < K < L-1$ layers while the adaptive part consists of the remaining $L - K$ layers. The adaptive component is a shared network that consists of sub-networks with varying widths, where the width of each sub-network is a fraction of the width of the network.

More formally, sub-networks are associated with a given a set of reduction factors $\{r_i\}_{i=1}^M, 0 < r_i \le 1$, where the width of $i$-th sub-network is $1/r_i \times$ smaller than that of main network. We map the hardness level $i$ to the width reduction factor $r_i$. During training, for each input, we sample one of the re-

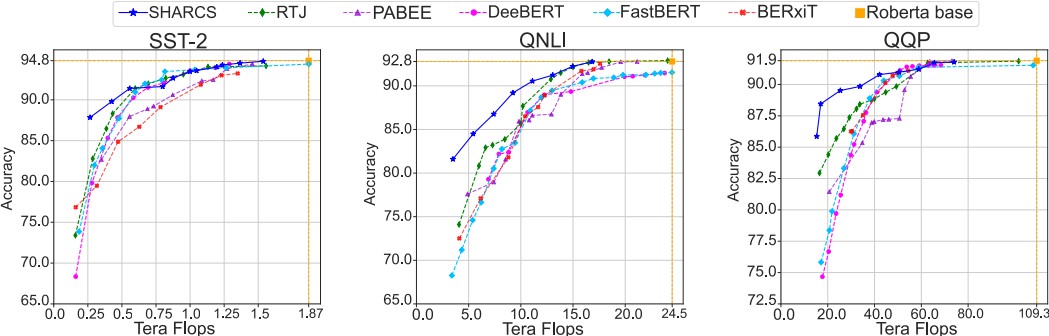

Figure 2: Results on the three of the sub-tasks in the GLUE benchmark. For the full set of plots on 8 tasks please refer to the appendix D.0.1. Best viewed in color.

duction factors with entry 1 in the router label for that input and do the forward and backward pass with just the network associated with that reduction factor. During inference, given the output of non-adaptive network for input sample $x$, the objective of the router is to determine the width of sub-network in the adaptive module to process $x$. We train the sub-networks and the router with the following objective:

$$\mathcal{L} = \lambda_{task} \cdot \mathcal{L}_{task} + \lambda_{router} \cdot \mathcal{L}_{router} \quad (1)$$

where $\mathcal{L}_{router}$ is a binary cross-entropy loss between predicted and ground-truth hardness label, $\mathcal{L}_{task}$ is a task-specific loss, and $\lambda_{task}$ and $\lambda_{router}$ are loss weights which are hyper-parameters.

## 2.3 Reducing Width of Adaptive Module

The basic building block in both multi-head attention (MHA) module and feed forward network (FFN) in transformers is the linear layer (Vaswani et al., 2017). Given the fully-connected nature of linear layers, to reduce their width, we retain the leftmost $r \cdot d_{\text{model}}$ neurons in both input and output (Kusupati et al., 2022). It is worth noting that as we reduce the input and output dimensions of matrix multiplications by a factor of $1/r$, the flops will be reduced by a factor of $1/r^2$. We follow a similar procedure for reducing the width of affine parameters in LayerNorm (Ba et al., 2016). Also in our setup, we do not change the head dimensions in MHA and instead decrease the number of heads by a factor of $1/r$. As Figure 1a illustrates, we down-project the input hidden states to the adaptive layers using a pooler module and up-project it back before feeding to the single classifier for all of the sub-networks. Please see Appendix B for a

detailed description of width reduction of different components in transformer layer.

## 3 Experimental setup

**Datasets.** We evaluate SHARCS on 8 classification tasks in the standard GLUE benchmark (Wang et al., 2018): MNLI-m (Williams et al., 2018), QNLI (Wang et al., 2018; Rajpurkar et al., 2016), QQP (Wang et al., 2017), SST-2 (Socher et al., 2013), MRPC (Dolan and Brockett, 2005), RTE (Dagan et al., 2005), CoLA (Warstadt et al., 2019), WNLI (Levesque et al., 2012).

**Evaluation metrics.** Following previous work, we report the accuracy on the validation set, with an exception to CoLA for which we report Matthews correlation. We measure the total FLOPs on the validation set as it is invariant to the run time environment (Liu et al., 2020).

**Training details.** We train network with AdamW optimizer (Loshchilov and Hutter, 2019). We choose the number of epochs in $\{5, 10, 15\}$ and use learning rate in $\{2e-5, 5e-5\}$ in our experiments. Please see Appendix C for more details.

## 4 Results

### 4.1 SHARCS is Better

We compare SHARCS with existing sample adaptive inference methods: RTJ (Schwartz et al., 2020), DeeBERT (Xin et al., 2020), FastBERT (Liu et al., 2020), BERxiT (Xin et al., 2021), and PABEE (Zhou et al., 2020). Figure 2 shows the accuracy-FLOPs trade-off plots for different sample adaptive approaches with RoBERTa_base model on three GLUE subtasks (See appendix D.0.1 for the full set of plots). Our results show that SHARCS significantly outperforms other methods, especially in

Table 1: **Comparison of different adaptive inference methods on MNLI-m dataset for different FLOP ranges**. For the full table, please see Appendix D.0.2

| FLOPs range (Tera FLOPs) | Best Baseline | | SHARCS | |
|---|---|---|---|---|
| | Acc.(%) ↑ | FLOPs ↓ | Acc. | FLOPs |
| **Roberta (Acc: 87.6, FLOPs: 33.5)** | | | | |
| 0-10 | 61.83 | 9.56 | **76.37** | **4.91** |
| 10-20 | 84.19 | 18 | **85.93** | **17.92** |
| 20-30 | **87.53** | 29.86 | 87.38 | **28.35** |
| **BERT (Acc: 84.8, FLOPs: 33)** | | | | |
| 0-10 | 48.87 | **8.76** | **72.7** | 8.94 |
| 10-20 | 72.50 | 19.99 | **81.61** | **16.73** |
| 20-30 | 83.27 | 27.99 | **83.04** | **22.76** |
| **DistilBERT (Acc: 82.2, FLOPs: 16.57)** | | | | |
| 0-5 | 48.87 | 4.7 | **64.72** | **3.80** |
| 5-10 | 64.44 | 9.95 | **76.02** | **8.87** |
| 10-15 | 80.38 | 14.91 | **81.61** | **14.39** |
| **DynaBERT 0.25 width (Acc: 83.9, FLOPs: 8.26)** | | | | |
| 0-4 | 65.61 | **3.78** | **78.92** | 3.99 |
| 4-6 | 76.01 | 5.42 | **81.48** | **5.21** |
| 6-8 | **83.86** | 7.92 | 83.40 | **7.33** |

the low-FLOPs regime. This can suggest that by substantially reducing the width of deeper layers in the model, SHARCS can achieve a significant reduction in FLOPs. Furthermore, we can maintain the accuracy better compared to fully skipping deeper layers as commonly done in early exiting methods.

## 4.2 SHARCS is Model Agnostic

SHARCS can be seamlessly integrated with any state-of-the-art non-efficient and efficient transformer-based encoders to further improve their efficiency. We use SHARCS and the baseline sample adaptive inference methods in (§4.1) with two standard non-efficient (RoBERTa$_{base}$ and BERT-Base$_{base}$) and efficient (DistilBERT (Sanh et al., 2019) and DynaBERT (Hou et al., 2020)) models. Table 1 shows that SHARCS consistently improves the efficiency of different models while maintaining the accuracy better than other baseline sample adaptive methods. Note that, for brevity, we have selected the highest accuracy among all the sample adaptive baselines for each FLOPs range; please see Table 5 in Appendix D.0.2 for the full set of results. As an example, SHARCS can reduce the FLOPs of BERT$_{base}$ to half with near 3% reduction in accuracy whereas in other methods the drop is more than 12%. Interestingly, SHARCS can further improve the efficiency of already optimized networks. For instance, inference with SHARCS on DynaBERT 0.25 width takes 10-15% less FLOPs with less than 1% drop in accuracy. More accuracy vs. FLOPs trade-off results for DynaBERT 0.25 and DistilBERT can be found in Figure 10 and 11.

## 4.3 SHARCS is Fast

We compare the latency of SHARCS applied to BERT$_{base}$ with the baseline adaptive approaches on the QQP (Wang et al., 2017) dataset. The latency measurements are conducted on an Intel Xeon Gold 6230 CPU with a batch size of 1. We use two reduction factors $\{0.25, 1.0\}$ and place the router after layer 2 or 4 to get different accuracy-speed up trade offs. We keep the speed up between 2 to 3 and report the best accuracy for each method in this range. Table 2 shows that our method obtains higher or comparable accuracy in this speed up range. Interestingly, SHARCS achieves a performance on par with DistilBERT, which is trained with distillation on a much larger dataset, but with higher speed up.

## 4.4 SHARCS Paired with Token Dropping

As discussed in Section 1, in addition to early-exiting, token dropping is another well-known approach to sample-adaptive inference efficiency (Goyal et al., 2020; Guan et al., 2022). Token dropping enhances efficiency by progressively dropping tokens layer by layer, thereby decreasing the hidden states sequence length as the forward pass proceeds. In contrast, SHARCS improves efficiency via reducing the transformer block's width. Therefore token dropping and SHARCS should not interfere with each other and in fact, they can be paired together to bring more efficiency. To showcase this, we combine SHARCS with a RoBERTa$_{base}$ network that has already been made more efficient using Transkimmer token dropping. Table 3 shows that on QNLI dataset, SHARCS reduces the FLOPs of RoBERTa + Transkimmer by 40% with a negligible drop in accuracy.

Table 2: **Inference speed up results on the QQP dataset on CPU for various adaptive inference techniques applied to BERT**. For SHARCS we place the router after layers 2 and 4 and report two numbers.

| | Accuracy (%) ↑ | Speed up (×) ↑ |
|---|---|---|
| BERT | 90.90 | 1 |
| DistilBERT | 88.50 | 2 |
| BERxiT | 85.60 | 2.04 |
| FastBERT | 83.99 | 2.17 |
| DeeBERT | 83.82 | 2.14 |
| PABEE | 89.09 | 2.06 |
| RTJ | 84.44 | 2 |
| **SHARCS (layer 2)** | 88.43 | **2.57** |
| **SHARCS (layer 4)** | **90.05** | 2.05 |

Table 3: **SHARCS can be applied to RoBERTa in addition to Transkimmer to bring more efficiency.** The decrease in FLOPs and Accuracy in parantheses are with respect to RoBERTa + Transkimmer.

|  | QNLI Acc. | TeraFLOPs |
|---|---|---|
| RoBERTa Base | 92.8% | 24.5 |
| +Transkimmer | 89.45% | 14.08 |
| **+Transkimmer + SHARCS** | 88.83% (-0.62%) | **8.56 ($\sim$40%$\downarrow$)** |

## 4.5 Ablating the Router

To show the efficacy of the router and its training strategy in SHARCS, we replace it with the routing strategy used in BERxiT (Xin et al., 2021): while training two sub-networks with reduction factors $\{0.25, 1.0\}$, we feed the training sample to both of them; if a sub-network correctly classifies the sample, the router label for that sub-network in that iteration would be 1. Otherwise, it would be zero. The backward pass is then done with all the losses of sub-networks and the router loss. Table 4 shows the area under the normalized accuracy FLOPs curve (AUC) for both methods on MNLI-m. Please find more detailed ablation results and experiments in Appendix E.

Table 4: While using adaptive width with RoBERTa$_{base}$ on MNLI-m, SHARCS router outperforms BERxiT router.

| Router on RoBERTa$_{base}$ | AUC $\uparrow$ |
|---|---|
| SHARCS | **0.78** |
| BERxiT | 0.73 |

## 5 Conclusion

We presented SHARCS as a new sample adaptive inference approach that can improve any network's inference efficiency. SHARCS incorporated a lightweight router which is trained with a novel approach using the confidence of the network predictions during training. Our experiments showed the superiority or complementary role of SHARCS compared to other sample adaptive inference methods across various datasets and backbones.

## Limitations

While the router and training approach in SHARCS are general purpose, a limitation of this paper is its focus solely on studying the impact on transformer encoders. Nonetheless, decoder-only (Radford et al., 2018, 2019) and encoder-decoder (Raffel et al., 2019; Lewis et al., 2020) models are widely used classes of models for which we plan to integrate SHARCS in the future work. It should be noted that although our approach can be applied to regression tasks with minor adjustments, this work does not include any results for regression tasks.

## Acknowledgements

We thank members of RAIVN and H2Lab for helpful discussion and feedback, and Hyak cluster team at the University of Washington for infrastructure support. This research was supported by NSF IIS-2044660, ONR MURI N00014- 18-1-2670, and gifts from AI2, Google and Apple.

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

## A  Background and Related Work

**Efficient non-adaptive inference**  In the context of non-adaptive methods, numerous studies in the literature have improved the inference efficiency of transformers through various techniques such as knowledge distillation (Hinton et al., 2015) into smaller models (Sanh et al., 2019; Sun et al., 2020; Hou et al., 2020; Jiao et al., 2019), pruning unimportant weights of the model (Xia et al., 2022; Lagunas et al., 2021; Sanh et al., 2020; Liu et al., 2019b), and weight quantization (Shen et al., 2019; Kim et al., 2021) to store weights of a network with lower precision values. The aforementioned approaches result in smaller and more efficient albeit fixed and static models.

**Efficient adaptive inference.**  Another line of work have proposed adaptive inference methods that allow the network to allocate varying amounts of compute for each sample. The predominant technique in this area is early exiting via adding internal classifiers to intermediate layers (Schwartz et al., 2020; Kaya and Dumitras, 2018; Xin et al., 2020; Teerapittayanon et al., 2017; Zhou et al., 2020; Liu et al., 2020): To early exit, prior work either use the confidence score of internal classifiers' predictions (Schwartz et al., 2020; Xin et al., 2020); the entropy of these predictions (Liu et al., 2020); a module that predicts whether a layer should early exit or not (Xin et al., 2021); a patience based change in these internal prediction (Zhou et al., 2020); or a hash based mechanism to do token level early exiting (Sun et al., 2022).

Devvrit et al. (2023) recently proposed MatFormer that can enable adaptive compute based on the resource constraints but does not utilize dynamic token-based routing making SHARCS complementary to it.

## B  Implementation Details

### B.0.1  Transformer Networks

Transformer networks (Vaswani et al., 2017) are composed of a stack of $L$ layers and each layer has two main components: Multi-head attention (MHA) and Feed-forward network (FFN).

**MHA**  consists of $n_h$ heads where each head computes the attention operation (Vaswani et al., 2017) on the projection of a sequence of $l$ tokens $x = (x_1, x_2, ..., x_l)$ into key, query, and values: $o_{\text{head}}^i = \text{Attention}(\mathbf{W}_K x, \mathbf{W}_\mathbf{Q} x, \mathbf{W}_\mathbf{V} x)$, wherein

$1 \leq i \leq H$, $o_{\text{head}}^i \in \mathbb{R}^{d_{\text{head}}}$ and $\mathbf{W}_K, \mathbf{W}_Q, \mathbf{W}_V \in \mathbb{R}^{d_{\text{model}} \times d_{\text{head}}}$ are the key, query, and value projection matrices, and $d_{\text{head}} = d_{\text{model}}/n_h$. The outputs from different heads are concatenated into $o_{\text{heads}} \in \mathbb{R}^{d_{\text{model}}}$ and projected with another matrix $\mathbf{W}_O \in \mathbb{R}^{d_{\text{model}} \times d_{\text{model}}}$ and passed through a layer norm (Ba et al., 2016) to get the output: $o_{\text{MHA}} = \text{LayerNorm}(x + \mathbf{W}_o o_{\text{heads}})$.

**FFN**  consists of two feed-forward layers $W_1$ and $W_2$ with GeLU (Hendrycks and Gimpel, 2016) non-linearity and takes the output of MHA module and computes $\text{LayerNorm}(o_{\text{MHA}} + \text{GeLU}(\mathbf{W}_1 o_{\text{MHA}})\mathbf{W}_2)$.

### B.0.2  Reducing Width of a Transformer Layer

Figure 3 illustrates our approach in reducing the capacity of a transformer network based on the given reduction factor. We leave the first $K$ layers (aka non-adaptive module) of the model intact, where $K$ is a hyperparameter. Given an input sequence $x$, after computing the intermediate representation $h \in \mathbb{R}^{l \times d_{\text{model}}}$ by the non-adaptive module, and the reduction factor $r$ by the router, we reduce the width of different components which we will describe in detail next.

Before passing $h$ to adaptive layers, we pass it through a pooler module that reduces its dimensionality to $d_{\text{model}} \times r$. For example, if $d_{model} = 768$ and $r = 0.25$, the input to the adaptive module will be 192 dimensional. Although we experimented with affine transformations for the pooler, in practice we observed that a simpler pooling operation such as selecting just the first $d_{model} \times r$ dimensions works as good. Therefore, as this pooler is simpler and importantly adds no extra FLOPs to inferece, we opted for that. For the unpooler, we used an affine transformation that transforms the hidden state with reduced dimensionality (i.e. $r \cdot d_{\text{model}}$) to $d_{model}$. Note that for the unpooler layer, the dimensionality of the input can change while the output dimensionality if always $d_{\text{model}}$.

Throughout this paper, whenever we decrease a dimension of a component (e.g. input vectors or weights) from $d$ to $d'$, we exclusively use the first $d'$ dimensions and disregards the remainder in the computations. This is illustrated schematically in Figure 4 for FFN and MHA components in transformer models.

In what follows we describe how we reduce the width of different components by the reduction

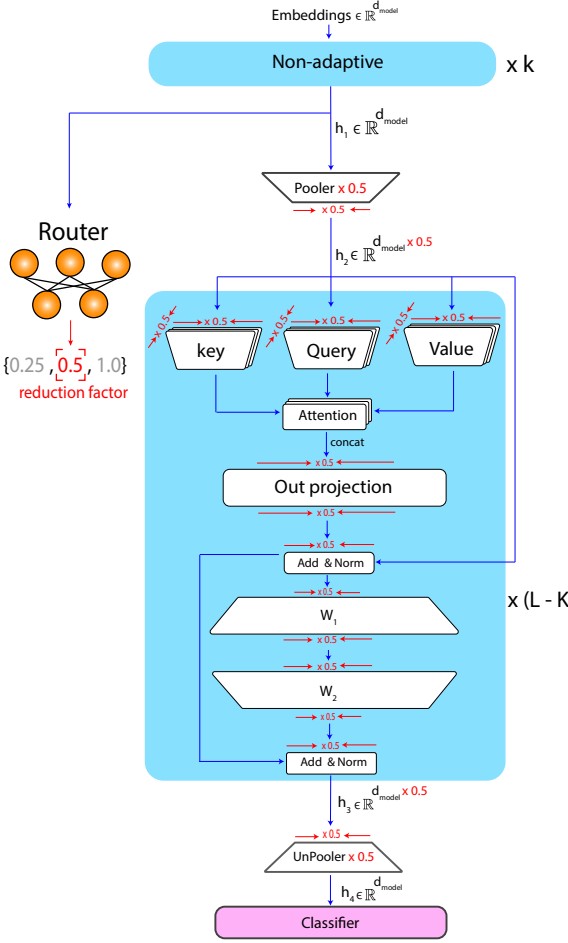

Figure 3: Detailed schematic of reducing with of different components in a transformer model based on the prediction of the router.

factor $r$ which is also depicted in detail in Figure 3.

**Reducing MHA Width.** We reduce the width of MHA using the following steps:

1. The input dimension $d_{model}$ of self-attention projection matrices $\mathbf{W}_K$, $\mathbf{W}_Q$, and $\mathbf{W}_v$ is decreased to $d_{model} \cdot r$. We do not change the output dimension $d_{head}$ of the linear projections.

2. The adaptive module only computes the output of the first $n_h \cdot r$ heads and disregards the other heads. Therefore the dimensionality of $o_{heads}$ is decreased to $d_{model} \cdot r$. Note that we could also reduce the dimensionality of each head instead, however, we built on previous findings in the literature that many of the heads in transformers are redundant and can be pruned without hurting the performance (Michel et al., 2019; Voita et al., 2019).

3. For the output projection $\mathbf{W}_o$ of MHA, the input and output dimensions, which are both equal to $d_{model}$, are reduced to $d_{model} \cdot r$.

Throughout all the changes above, we do not alter the sequence length of the input (i.e. $l$) or any of the hidden states.

**Reducing FFN width.** Similar to the output projection $\mathbf{W}_o$ of MHA, the input and output dimensions of the feed-forward layers $\mathbf{W}1$ and $\mathbf{W}2$ in the FFN are reduced by a factor of $1/r$. Therefore, the output dimension of all the adaptive layersare reduced by a factor of $r$ to $D_{model} \times r$.

It is important to note that majority of operations in FFN and MHA components are comprised of matrix multiplications and reducing the input and output dimensions of these multiplications by a factor of $r_j$ will reduce their flops by a factor of $r_j^2$.

**Reducing layernorms width.** We also reduce the width of the layer norm parameters by a factor of $r$. In our experiments, we find it beneficial to initialize the layernorm parameters with the corresponding first dimensions of original layernorm parameters instead of training them from scratch.

### B.0.3 Inference

In contrast to training where the width of the adaptive layers doing forward pass is enforced by the router labels, at inference, the router predicts the reduction factor. More formally, given the input sample $x$ and router logits $\hat{W} \in \mathbb{R}^M$, we select the reduction factor $r_j$ to do the forward pass where j is $\arg\max(\hat{W})$.

## C  Training Details

Similar to (Hou et al., 2020), we reorder the heads based on their importance before fine-tuning on the down-stream task. We set $\lambda_{task}$ to 1 and $\lambda_{router}$ to 0.5 in our approach and use a set of two reduction factors $\{0.25, 1.0\}$. We choose the batch size in $\{16, 32\}$ depending on the model and dataset. We do a grid search over the lower and upper bound on the confidence thresholds and do the forward pass of each reduction factor on a separate GPU. We choose the window size values in $\{3, 5\}$. We set $\lambda_{task}$ to 1 and $\lambda_{router}$ to 0.5 in our approach and use a set of two reduction factors $\{0.25, 1.0\}$. We do a grid search over the lower and upper bound on the confidence thresholds and the hyperparamters of the baselines and do the forward pass of each reduction factor on a separate GPU. Similar to previous work (Hou et al., 2020), we pad the input sequences in a batch to 128 tokens while training. To train SHARCS with Transkimer on RoBERTa, we first train RoBERTa + Transkimmer with skim coefficient of 0.3. We then start from this model as

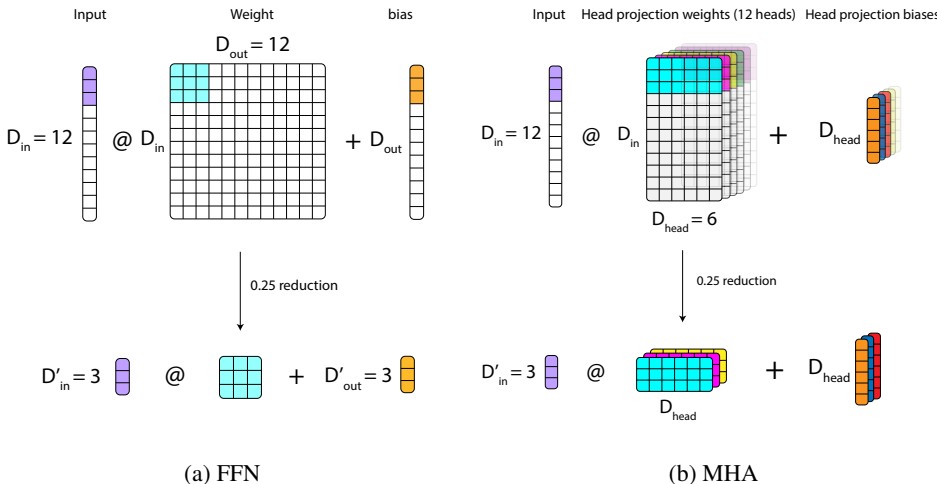

(a) FFN                                    (b) MHA

Figure 4: Reducing the width of FFN and MHA in adaptive layers.

an initial checkpoint and train for 10 epochs using both our loss and Transkimmer loss. Here, training the router is done with with 0.25 and 1.0 as reduction factors.

## D Results Details

### D.0.1 GLUE Results

We do a thorough hyperparameter search with different sample adaptive inference methods to get the accuracy of the RoBERTa$_{base}$ model across different FLOPs values. We report the total number of FLOPs on the validation set and do not pad the input sequences before feeding to the model. Figure 12 shows the plots for 8 of the sub-tasks in the GLUE benchmark.

### D.0.2 Different Backbone Results

Table 5 shows the comparison of different sample adaptive inference methods applied to different backbones.

## E Ablations and Discussion

### E.0.1 Router Ablation

Figure 5 shows the accuracy FLOPs plot for two methods: 1) The router introduced in SHARCS with adaptive width sub-networks. 2) A router similar to the early-exiting router used in BERxiT (Xin et al., 2021) with adaptive-width sub-networks. We report the AUC for a similar range of FLOPs for both methods and scale the accuracy and FLOPs values to [0, 1].

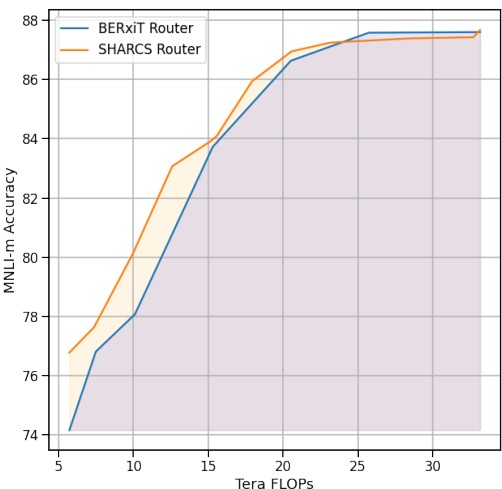

Figure 5: Accuracy vs. FLOPs plot for Adaptive width sub-networks trained with SHARCS router (orange) and BERxiT (Xin et al., 2021) router (blue).

### E.0.2 Number of Reduction Factors

We change the number of reduction factors (or hardness levels) $M$ to three and four and use the reduction factors $\{0.25, 0.5, 1.0\}$ and $\{0.25, 0.5, 0.75, 1.0\}$ respectively. Figure 6 shows the results of changing the number of reduction factors with RoBERTa$_{base}$ model on MNLI-m dataset. The model trained with two reduction factors outperform the other cases for FLOPs values above 15 teraFLOPs significantly and four reduction factors can get better results in the range of 10-15 Tera FLOPs.

Table 5: Comparison of different adaptive inference methods on MNLI-m dataset for different FLOP ranges.

| FLOPs range (Tera Flops) | RTJ | | DeeBERT | | PABEE | | FastBERT | | BERxiT | | SHARCS | |
|---|---|---|---|---|---|---|---|---|---|---|---|---|
| | Acc. ↑ | FLOPs ↓ | Acc. | FLOPs | Acc. | FLOPs | Acc. | FLOPs | Acc. | FLOPs | Acc. | FLOPs |
| **Roberta (Acc: 87.6, FLOPs: 33.5)** | | | | | | | | | | | | |
| 0-10 | 54.27 | 9.59 | 61.83 | 9.56 | 53.98 | 8.53 | 45.39 | 2.84 | 53.69 | 8.41 | **76.37** | **4.91** |
| 10-20 | 82.29 | 19.77 | 77.18 | 19.80 | 84.19 | 18 | 59.79 | 15.06 | 83.39 | 19.91 | **85.93** | **17.92** |
| 20-30 | 86.92 | 25.76 | 81.35 | 23.67 | 87.53 | 29.86 | 77.24 | 29.95 | 87.23 | 27.92 | **87.38** | **28.35** |
| **BERT (Acc: 84.8, FLOPs: 33)** | | | | | | | | | | | | |
| 0-10 | 47.32 | 9.46 | 42.32 | 9.97 | 42.97 | 7.51 | 48.87 | 8.76 | 46.86 | 9.88 | **72.7** | **8.94** |
| 10-20 | 58.12 | 15.61 | 61.57 | 19.89 | 66.33 | 17.29 | 70.25 | 19.18 | 72.50 | 19.99 | **81.61** | **16.73** |
| 20-30 | 78.96 | 28.53 | 80.57 | 29.34 | 81.13 | 27.66 | 83.27 | 27.99 | 80.82 | 25.73 | **83.04** | **22.76** |
| **DistilBERT (Acc: 82.2, FLOPs: 16.57)** | | | | | | | | | | | | |
| 0-5 | 41.92 | 2.76 | 43.98 | 2.76 | 41.45 | 2.76 | 48.87 | 4.7 | 41.47 | 4.07 | **64.72** | **3.80** |
| 5-10 | 52.72 | 9.56 | 59.87 | 9.45 | 64.44 | 9.95 | 64.24 | 9.76 | 55.10 | 9.76 | **76.02** | **8.87** |
| 10-15 | 72.93 | 13.90 | 80.38 | 14.91 | 79.29 | 14.12 | 79.72 | 14.84 | 60.25 | 11.68 | **81.61** | **14.39** |
| **DynaBERT 0.25 width (Acc: 83.9, FLOPs: 8.26)** | | | | | | | | | | | | |
| 0-4 | 60.95 | 3.98 | 56.13 | 3.38 | 56.09 | 3.97 | 65.61 | 3.78 | 56.19 | 3.58 | **78.92** | **3.99** |
| 4-6 | 74.55 | 5.82 | 75.26 | 5.62 | 69.81 | 5.93 | 76.01 | 5.42 | - | - | **81.48** | **5.21** |
| 6-8 | 83.43 | 7.70 | 83.36 | 7.70 | 81.31 | 7.87 | 83.86 | 7.92 | 83.08 | 7.60 | **83.40** | **7.33** |

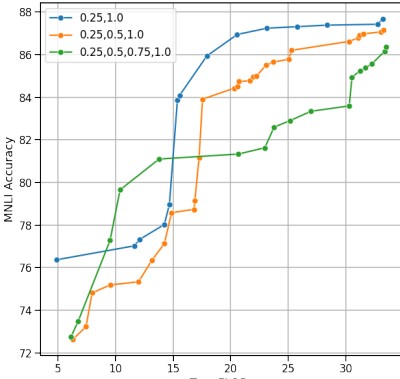

Figure 6: Results of different number of sub-networks (or reduction factors) with RoBERTa$_{base}$ model on MNLI-m.

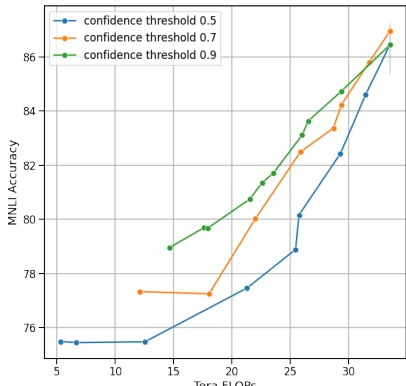

Figure 7: Results of changing confidence thresholds with RoBERTa$_{base}$ model on MNLI-m dataset. We use reduction factors $\{0.25, 1.0\}$ and confidence thresholds $[0, x]$ for reduction factor 0.25 and confidence thresholds $[x, 1]$ for reduction factor 1, where $x \in \{0.5, 0.7.0.9\}$.

### E.0.3  Confidence Thresholds

We study the effect of changing confidence thresholds on our results with a simple experiment: With two reduction factors $\{0.25, 1.0\}$ we use the following confidence score lower and upper bounds: $[0.0, x]$ for full network and $[x, 1.0]$ for 0.25 network, where $x \in \{0.5, 0.7, 0.9\}$. We place the router after layer 1 (i.e. $K = 1$). Figure 7 shows the results of RoBERTa$_{base}$ model on MNLI-m dataset. According to the figure, higher value of $x$ leads to better acuracies for a fixed number of FLOPs. Furthermore, as we decrease $x$, we can reach lower values of FLOPs. This is intuitive as decreasing the lower bound of smallest reduction factor makes more samples be routed to that which reduces the overall FLOPs.

### E.0.4  Using Entropy or Confidence-based Hardness Labels to Train Router

Similar to sample adaptive inference in early exiting methods (Xin et al., 2020; Liu et al., 2020), one can get hardness labels by defining thresholds on entropy of the network predictions instead of the confidence. We use the following formula to compute the entropy of network's prediction:

$$H = -\sum_{i=1}^{C} p_i \cdot \log p_i,$$

wherein $C$ denotes the number of classes of the classifier, and $p_i$ is the softamx probablity that the

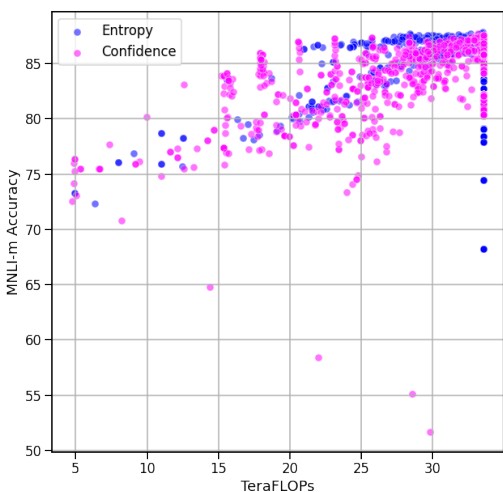

Figure 8: Comparing entropy and confidence based hardness labels for training the router.

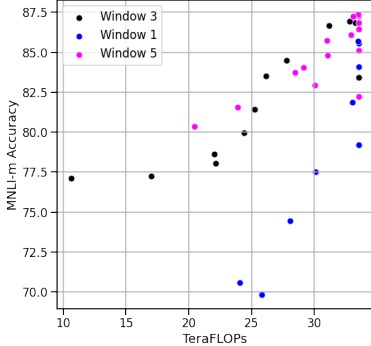

Figure 9: Effect of changing history window size in SHARCS on MNLI-m Accuracy FLOPs trade-off.

classifier assigns to class $i$. Figure 8 shows the accuracy FLOPs trade-off for both of these metrics on MNLI-m dataset with RoBERTa$_{base}$ model. Given that the confidence score is a simpler approach and does not require any additional computation, we opted for using that in our method.

### E.0.5 Changing History Window Size in Training the Router

As mentioned in (§2.1), having a larger window size helps stabilizing training the router as the hardness label for a training sample might change throughout training. To illustrate this effect, we train SHARCS on RoBERTa$_{base}$ model with three different window sizes $\{1, 3, 5\}$ for 10 epochs on MNLI-m dataset. We place the router at layer 2, use two reduction factors $\{0.25, 1.0\}$, and set the

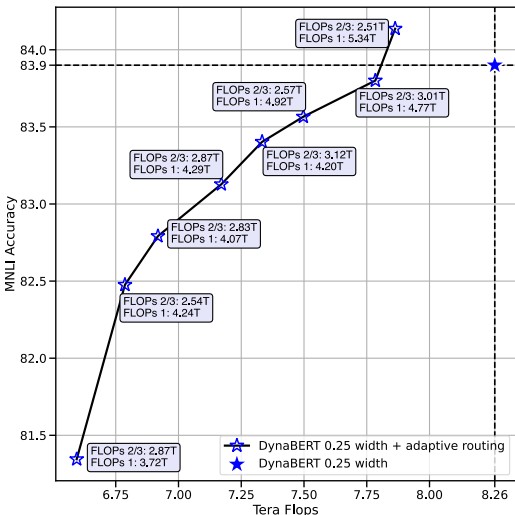

Figure 10: Accuracy FLOPs plot for SHARCS applied to DynaBERT 0.25 W (Hou et al., 2020) model. We set the reduction factors to $\{2/3, 1.0\}$. By placing router at different layers and changing the confidence thresholds, we can get different points in the plot. Note that the FLOPs for each sub-network in each point is also reported.

confidence thresholds to $[0.0, 0.8]$ for reduction factor 1 and $[0.8, 1.0]$ for reduction factor 0.25. Figure 9 shows the accuracy FLOPs trade offs that different checkpoints with different window sizes get throughout training on the validation set. According to the figure, the network can reach higher accuracy when trained with window size 3 or 5. We did not see any significant improvements by using a window size larger than 5.

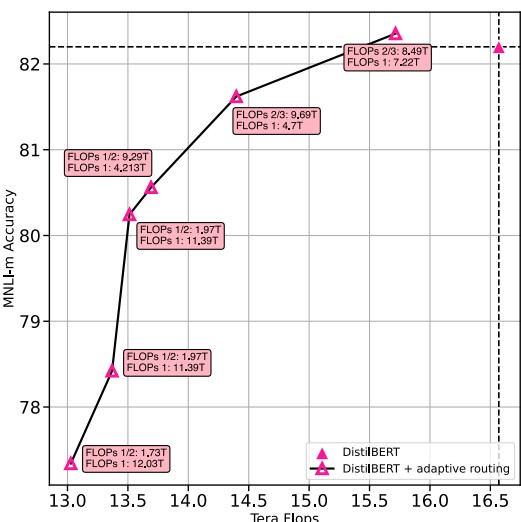

Figure 11: Accuracy FLOPs plot for SHARCS applied to DistilBERT(Sanh et al., 2019) model. We set the reduction factors to $\{2/3, 1.0\}$ or $\{0.5, 1.0\}$. By placing router at different layers and changing the confidence thresholds, we can get different points in the plot. Note that the FLOPs for each sub-network in each point is also reported.

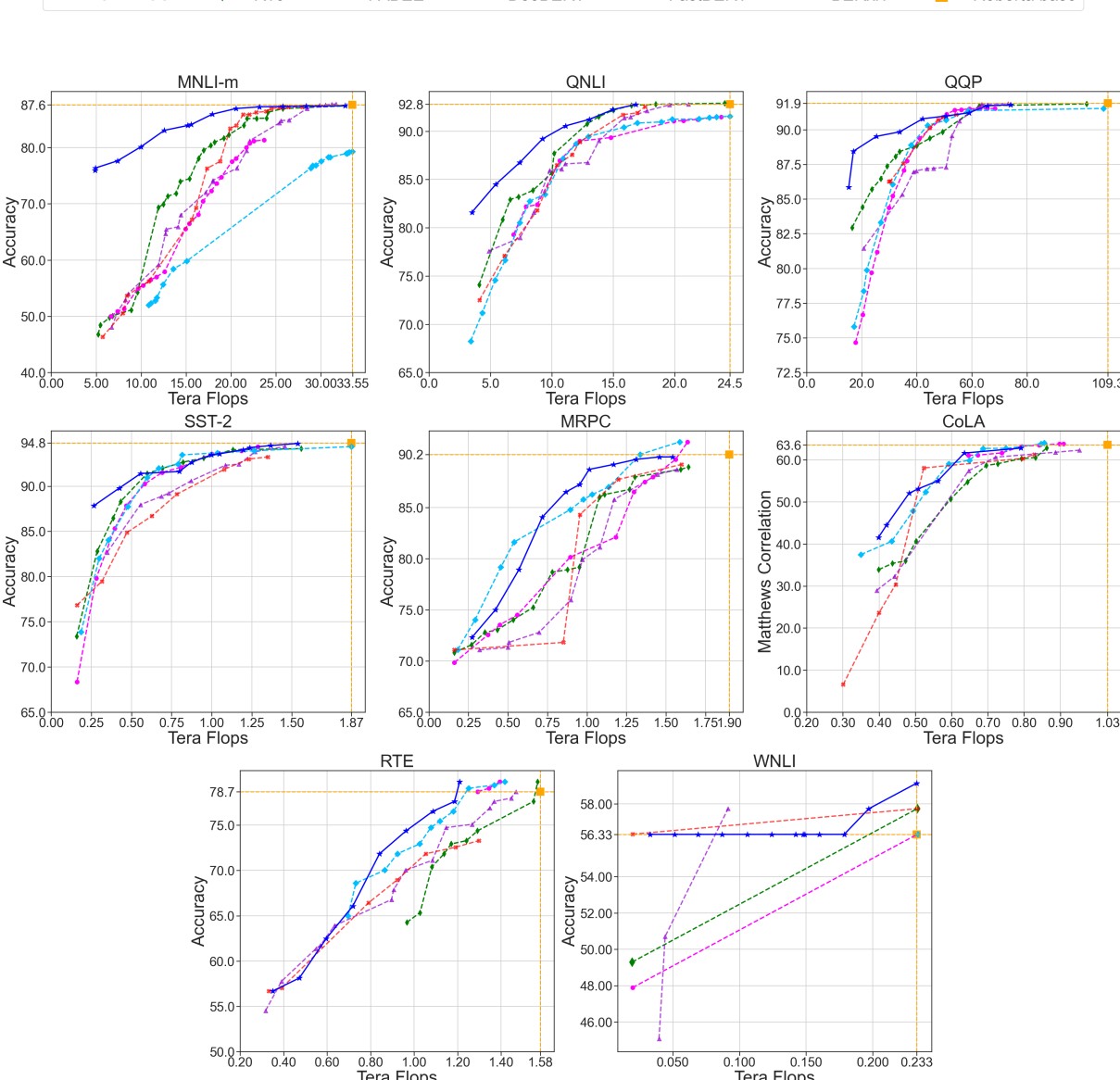

Figure 12: GLUE benchmark results. Best viewed in color.