# OpenReview forum: "SHARCS: Efficient Transformers Through Routing with Dynamic Width Sub-networks"
_EMNLP/2023/Conference — EMNLP 2023 Findings_

### Official Review · Reviewer_cRFX · 2023-07-19

**Soundness:** 3

**Excitement:**

3: Ambivalent: It has merits (e.g., it reports state-of-the-art results, the idea is nice), but there are key weaknesses (e.g., it describes incremental work), and it can significantly benefit from another round of revision. However, I won't object to accepting it if my co-reviewers champion it.

**Paper Topic And Main Contributions:**

This paper proposes a method, SHARCS, to accelerate the inference of Transformer-based models by introducing a router. To train the router, authors first design a heuristical method to estimate the hardness of samples, then finetune the sub-networks and router, simultaneously. The experiments show that SHARCS outperforms prior methods and improves the inference efficiency.

**Questions For The Authors:**

Q1: As shown in Eq. (1) and Line156, the router only is trained by the binary cross-entropy loss, which does not involve information about the degree of hardness of samples. So, why does it have discriminative ability (i.e. the router can assign the reduction factor based on the hardness level of a sample) during inference?

Q2: When making inference, the hardness of the samples within the same batch may be quite different, and therefore the reduction factor assigned should be diverse. In this case, how to perform the parallel computation?

**Reasons To Accept:**

- The paper is well organized and easy to follow.
- The paper presents the approach that addresses an important problem (i.e. efficient inference) in application.
- The paper conducts extensive experiments to demonstrate the effectiveness of the proposed method.

**Reasons To Reject:**

- Poor writing quality: although this paper is easy to understand, a thorough proofreading of the paper is still required (e.g., expression).

- Lack of novelty: the paper presents a method that is not sufficiently novel or innovative, and the paper fails to provide a clear and compelling rationale for why the proposed method is necessary and how it differs from existing methods.

- Insufficient experimental comparison: (i) the comparison results on the test set are more convincing than those on the validation set; (ii) ablation experiments are necessary on the number of non-adaptive layers (i.e. K).

**Reproducibility:**

4: Could mostly reproduce the results, but there may be some variation because of sample variance or minor variations in their interpretation of the protocol or method.

**Reviewer Confidence:**

3: Pretty sure, but there's a chance I missed something. Although I have a good feel for this area in general, I did not carefully check the paper's details, e.g., the math, experimental design, or novelty.

---

> ### Author Rebuttal · Authors · 2023-08-29
>
> We would like to thank the reviewer for their time and valuable feedback. Please read our comment on positioning our paper in the literature which is available [here](https://openreview.net/forum?id=M1GRz46Ahz&noteId=GlysqKUbZV).
>
> We address the concerns and questions of the reviewer as follows:
>
> **Poor writing quality: although this paper is easy to understand, a thorough proofreading of the paper is still required (e.g., expression).**
>
> We are glad the reviewer found the paper easy to follow and understand. We shall proof-read the paper further to improve the writing; however, we would be grateful if the reviewer could point us to the specific parts that require proof-reading so that we can improve it.
>
> **Lack of novelty: the paper presents a method that is not sufficiently novel or innovative, and the paper fails to provide a clear and compelling rationale for why the proposed method is necessary and how it differs from existing methods.**
>
> We believe that there are a few important reasons why our approach outperforms the previous methods:
> 1. SHARCS router is effective in learning the hardness of input samples. This allows samples to be processed with appropriate amounts of compute which in turn results in inference efficiency. The ablation at section 4.4 line 257 in the paper shows that replacing SHARCS router with a router that is used in an early exiting methods result in worse accuracy FLOPs trade-off.
> 2. The accuracy drops are more steep in adaptive depth methods compared to adaptive width. This is mainly due to the fact that skipping an entire layer can significantly hurt the performance. In contrast, our method does not fully skip deeper layers; rather it substantially improves their efficiency by decreasing input and output dimensions of all projection matrices. For example, the first projection weight of the FFN component in each layer of BERT has dimensions of
> $ 768 * 3072 $.
> In a $0.25$-width sub-network in SHARCS, these dimensions will be decreased to $ 192 * 768 $. Therefore the projection computation requires 16 times less FLOPs. This also holds true for other components in the transformer layer as they are all essentially projection layers.
> 3. While SHARCS is effective in reducing the FLOPs by a large factor, table 2 of the paper shows that it can also result in a significant speed up. In fact, the substantial reduction in the FLOPs translates to better latency – although not 16 times faster, as FLOPs and latency are not linearly correlated.
>
> **Insufficient experimental comparison: (i) the comparison results on the test set are more convincing than those on the validation set; (ii) ablation experiments are necessary on the number of non-adaptive layers (i.e. K).**
>
> (i) As the labels for GLUE test set are not publicly accessible, following  most existing work that evaluate on GLUE dataset, we do not report any numbers on the test set and use the validation set as our test set.
>
> (ii) Regarding ablations, we have done ablations on number of sub-networks (appendix E.0.2 line 682), confidence thresholds (appendix E.0.3 line 694), using confidence or entropy as the measure of hardness (appendix E.0.4 line 710) and the window size during training (appendix E.0.5 line 721). We also did an extra ablation on $k$ on MNLI dataset for the rebuttal (i.e. number of non-adaptive layers) and the following table shows the results:
>
> | $k$ (# of non-adaptive layers) | MNLI-m Acc. | TeraFLOPs |
> |----------|----------|----------|
> | $1$    | $72.35$   | $6.35$ |
> | $3$   |  $79.95$  | $16.31$   |
> | $5$   |  $83.67$  | $18.71$  |
> | $7$   |  $86.51$  | $21.67$  |
> | $9$   |  $86.59$  | $26.40$   |
>
> The table shows that having more non-adaptive layers lead to better accuracy yet at the cost of higher FLOPs.
>
> **Q1: As shown in Eq. (1) and Line156, the router only is trained by the binary cross-entropy loss, which does not involve information about the degree of hardness of samples. So, why does it have discriminative ability (i.e. the router can assign the reduction factor based on the hardness level of a sample) during inference?**
>
> To make the sub-networks robust to router’s noise, we allow the hardness labels to be multi-hot labels which makes training the router a multi-label classification problem. During inference the reduction factor that has been assigned the highest logit by the router is selected and will be applied to the adaptive part. This will induce the necessary discriminative ability.
>
> **Q2: When making inference, the hardness of the samples within the same batch may be quite different, and therefore the reduction factor assigned should be diverse. In this case, how to perform the parallel computation?**
>
> It is important to note that in many user-centric applications (e.g. models on edge devices or streaming settings) batch size of 1 is often used. In case of models hosted on cloud, if the batch size is large enough (e.g. 256 or larger), there shall be enough examples for each reduction factor to run adaptive layers for them in parallel through batching.

---

### Official Review · Reviewer_gQht · 2023-08-05

**Soundness:** 3

**Excitement:**

3: Ambivalent: It has merits (e.g., it reports state-of-the-art results, the idea is nice), but there are key weaknesses (e.g., it describes incremental work), and it can significantly benefit from another round of revision. However, I won't object to accepting it if my co-reviewers champion it.

**Missing References:**

Please refer to the relation to the existing width-wise compression methods (token removing approach in Reasons to Reject [1-3])

**Paper Topic And Main Contributions:**

The paper considers inference inefficiency in current pre-trained transformer models. To this end, the authors propose SHARCS, a width-adaptive models that first identifies the level of difficulty of the samples and then assigns them into different layers with varying widths. To realize such mechanism, the paper introduces a trainable router that predicts the level of difficulty based on the moving average of the confidence. Evaluation results show that the proposed mechanism show better speed-up compared to existing method while achieving comparable accuracy.

In the reviewer's viewpoint, the contributions of the paper are: (i) a new mechanism for efficient inference method (ii) better performance on wide-range of NLP tasks.


**Questions For The Authors:**

Q1) In Table 2, why SHARCS (layer-2) shows better speedup with better accuracy than SHARCS (layer-4)?

Q2) The labels for the router training are based on the confidence of the training models for the  training samples. However, as the training proceeds, the trained models tend to reveal over-confidence and low calibration, which can assign the most examples into the smallest subnetwork . How did the authors deal with such over-confidence? Moreover, it is great to see the real-examples about which samples are assigned to which groups.

Q3) Which factors makes the proposed method special or achieve better performance compared to existing methods?Can you explain the rationale behind the proposed method?

Q4) Why does the SHARCS use the leftmost dimensions for the reduction? How sensitive is the model when selecting rightmost or any directions?

**Reasons To Accept:**

The strengths of this paper are as follows:

(i) The proposed method is a new and has quite different perspective compared to existing early exiting methods, which can facilitate future research in NLP community.

(ii) Outperformed results on various NLP tasks and backbones.

**Reasons To Reject:**

The weaknesses of this paper are as follows:

(i) The outperformed results (in terms of speedup) can be seen only when the accuracy is significantly decreased (trade-off plots in Figure 2).

(ii) As far as I understand, the proposed method is not the early-exiting approach. However, the paper mainly compares SHARCS with early-exiting methods. In the field of inference efficiency, there are a number of active studies that reduce the width (in this case, the number of tokens processed within the transformer), please refer to [1-3]. Moreover, the paper compare the outdated static compression method (distillbert, proposed in 2019).

(iii) The last thing is the lack of 'why' part. Why does the proposed method reveal better speedup than existing early exit methods?

[1] PoWER-BERT: Accelerating BERT Inference via Progressive Word-vector Elimination, ICML'20

[2] AdapLeR: Speeding up Inference by Adaptive Length Reduction, ACL'22

[3] Transkimmer: Transformer Learns to Layer-wise Skim, ACL'22

**Reproducibility:**

3: Could reproduce the results with some difficulty. The settings of parameters are underspecified or subjectively determined; the training/evaluation data are not widely available.

**Reviewer Confidence:**

4: Quite sure. I tried to check the important points carefully. It's unlikely, though conceivable, that I missed something that should affect my ratings.

---

> ### Author Rebuttal · Authors · 2023-08-29
>
> We would like to thank the reviewer for their time and valuable feedback. Please read our comment on positioning our paper in the literature which is available [here](https://openreview.net/forum?id=M1GRz46Ahz&noteId=GlysqKUbZV).
>
> We address the concerns and questions of the reviewer as follows:
>
>
> **(i) The outperformed results (in terms of speedup) can be seen only when the accuracy is significantly decreased (trade-off plots in Figure 2).**
>
> We disagree that the SHARCS only outperform when the accuracy is significantly decreased. For example in MNLI-m and QNLI plots in figure 12, our method outperforms other methods even in the range of 0-5% accuracy drop of the base model. Furthermore, while we understand that the high compute regime is important, the performance should drop gracefully in the low-flops regime. This is especially important for low-resource scenarios wherein the compute constraints are harsh and the model needs to serve in a restricted setting.
>
> **(ii) As far as I understand, the proposed method is not the early-exiting approach. However, the paper mainly compares SHARCS with early-exiting methods. In the field of inference efficiency, there are a number of active studies that reduce the width (in this case, the number of tokens processed within the transformer), please refer to [1-3]. Moreover, the paper compare the outdated static compression method (distillbert, proposed in 2019).**
>
> It is important to note that we are introducing a new class of efficient models which leverages **adaptive width** and a **novel sample hardness-based router**. We do not alter the sequence length. This makes our approach complementary to existing approaches such as token reduction methods. To demonstrate this, we combined SHARCS with the Transkimer [4].
>
> To get the results we used Transkimer’s code and trained a model with skim coefficient 0.3. We then start from this model as initial checkpoint and train for 10 epochs using both our loss and Transkimer loss. We used two reduction factors 0.25 and 1.0. The table below shows that using SHARCS with Transkimer can reduce the FLOPS by near 40% while the accuracy drops only ~0.6%.
>
> | Model | QNLI Acc. | TeraFLOPs |
> |----------|----------|----------|
> | Roberta Base    | $92.8$   | $24.5$   |
> | Transkimer    | $89.45$   | $14.08$   |
> | **Transkimer + SHARCS**   | $88.83$ **(-0.62%)**   |  **$8.56$ (40% FLOPs reduction)**  |
>
>
> **(iii) The last thing is the lack of 'why' part. Why does the proposed method reveal better speedup than existing early exit methods?**
>
> We believe that there are a few important reasons why our approach outperforms the previous methods:
> 1. SHARCS router is effective in learning the hardness of input samples. This allows input samples to be processed with appropriate amounts of compute which in turn results in inference efficiency. The ablation at section 4.4 line 257 in the paper shows that replacing SHARCS router with a router that is used in an early exiting methods result in worse accuracy FLOPs trade-off.
> 2. The accuracy drops are more steep in adaptive depth methods compared to adaptive width. This is mainly due to the fact that skipping an entire layer can significantly hurt the performance. In contrast, our method does not fully skip deeper layers; rather it substantially improves their efficiency by decreasing input and output dimensions of all projection matrices. For example, the first projection weight of the FFN component in each layer of BERT has dimensions of $768 * 3072$. In a $0.25$-width sub-network in SHARCS, these dimensions will be decreased to $ 192 * 768 $. Therefore the projection computation requires $16$ times less FLOPs. This also holds true for other components in the transformer layer as they are all essentially projection layers.
> 3. While SHARCS is effective in reducing the FLOPs by a large factor, table 2 of the paper shows that it can also result in a significant speed up. In fact, the substantial reduction in the FLOPs translates to better latency – although not 16 times faster, as FLOPs and latency are not linearly correlated.
>
> **Q1) In Table 2, why SHARCS (layer-2) shows better speedup with better accuracy than SHARCS (layer-4)?**
>
> This is an oversight and the labels should be flipped. Thanks for pointing this out.
>
> **Q2) The labels for the router training are based on the confidence of the training models for the training samples. However, as the training proceeds, the trained models tend to reveal over-confidence and low calibration, which can assign the most examples into the smallest subnetwork . How did the authors deal with such over-confidence? Moreover, it is great to see the real-examples about which samples are assigned to which groups.**
>
> We believe that If the networks are trained for too many epochs then this problem can happen. However by setting the hyperparameters (e.g. learning rate and its scheduler and number of epochs) and thresholds correctly, we observed that this can be prevented. Let’s assume a scenario wherein window size is W and there are two sub-networks: 0.25 width and full width. Here is the training dynamics we observed in most of our experiments:
> 1. During the first window epochs, only the full width network is trained and confidence scores are recorded.
> 2. Before epoch W+1, a large percentage of the samples will be assigned to the smallest sub-network. At this point we start training both sub-networks. Many of the samples assigned to the smallest sub-network start bouncing back to the full width-network as they are too hard for this slimmest sub-network.
> 3. After training for a reasonable number of epochs by using early stopping, in many of our experiments we observed that near 60% of the samples were routed to the smallest sub-network with a minimal drop in accuracy compared to the original model.
>
> **Q3) Which factors makes the proposed method special or achieve better performance compared to existing methods?Can you explain the rationale behind the proposed method?**
> Please refer to our response to point **(iii)** and our general comment (available [here](https://openreview.net/forum?id=M1GRz46Ahz&noteId=GlysqKUbZV)) which describes our rationale for our adaptive width method and sample hardness-based router.
>
>
> **Q4) Why does the SHARCS use the leftmost dimensions for the reduction? How sensitive is the model when selecting rightmost or any directions?**
>
> Pooler can be any module that reduces the dimensionality of the hidden state by the chosen reduction factor. Here are the main pooler modules one can consider:
>
> 1. Taking the first $ r * d $ dimensions of the input where $r$ is the reduction factor and $d$ is model dimension (e.g. if
> $ r=0.25 $
> and $d=768$, we take the first $ 768 * 0.25 = 192 $ dimensions of the hidden state as the input to the adaptive component). This is the approach we take in the paper.
>
> 2. Using some other segment from the input. For example, the last dimensions or any random subset of dimensions in the input. We expect that this approach gets the same accuracy as the first approach. Previous related work in the literature observed that information is uniformly diffused in different segments of the hidden states whether they are drawn from the last, random, or initial dimensions [1, 2]. This is also intuitive as there is no inductive bias in the learning process that make different dimensions encode information differently and the model is not aware of the order of dimensions.
>
> 3. Projecting the input into a vector with a smaller dimensionality using a linear layer. We opt this one out as it adds to the computational complexity which goes against our goal of maintaining computational efficiency within the model.
>
> 4. One other option that we considered while designing the pooler was splitting the hidden state vector into sub-vectors with equal sizes and compute a weighted sum of them. For example, if the reduction factor is $0.25$ and model dimension is $768$, we chunk the hidden states into four $192$-d vectors $v_1$ to $v_4$ and take the sum $ w_1 . v_1 + … + w_4.v_4 $ where $ (w_1, w_2, w_3, w_4) $ is the weight parameter. This will only add four parameters to the method which is much more efficient than using a linear layer. The following table shows the results of comparison between SHARCS that uses first $ r * d$ dimensions and SHARCS that takes a weighted sum of chunks of hidden state. We average the results for threshold values from $0.1$ to $0.9$ for Roberta Base on QNLI while placing router after the third layer
> | Method | QNLI Avg. Acc. |  TeraFLOPs|
> |----------|----------|----------|
> | SHARCS (weighted sum of chunks)    | $82.09$   | $10.41$  |
> | SHARCS (first dimensions)    | $82.45$   | $9.97$  |
> | Roberta base   | $92.8$   | $24.5$ |
>
> The table suggests that SHARCS that uses the initial dimensions outperforms SHARCS that takes the weighted sum of the chunks in terms of accuracy FLOPs trade-off.
>
> Note that the unpooler is the linear layer before the classifier layer. In fact, it is common to have one layer which operates on CLS token of BERT based models before the classifier layer [3]. We reduce the input dimension of this pre-existing layer according to the chosen reduction factor and do not change the output dimension (i.e. the output dimension remains 768 which is the dimensionality of base models in the BERT family).
>
> [1] Kusupati, A., Bhatt, G., Rege, A., Wallingford, M., Sinha, A., Ramanujan, V., Howard-Snyder, W., Chen, K., Kakade, S., Jain, P., & others (2022). Matryoshka Representation Learning. In Advances in Neural Information Processing Systems.
>
> [2] Daniel Soudry, Elad Hoffer, Mor Shpigel Nacson, Suriya Gunasekar, & Nathan Srebro. (2022). The Implicit Bias of Gradient Descent on Separable Data.
>
> [3] Jacob Devlin, Ming-Wei Chang, Kenton Lee, & Kristina Toutanova (2018). BERT: Pre-training of Deep Bidirectional Transformers for Language Understanding.
>
> [4] Guan, Y., Li, Z., Leng, J., Lin, Z., & Guo, M. (2022). Transkimmer: Transformer Learns to Layer-wise Skim.

---

### Official Review · Reviewer_sLft · 2023-08-05

**Soundness:** 2

**Excitement:**

2: Mediocre: This paper makes marginal contributions (vs non-contemporaneous work), so I would rather not see it in the conference.

**Paper Topic And Main Contributions:**

This paper addresses the problem of a high computational cost of pre-trained language models. To reduce the cost, the authors propose a dynamic width transformer that routes subnetworks with different lengths based on the hardness of inputs. The experimental results in the paper show that the proposed method shows higher accuracy with commensurate cost compared with various compression methods.

**Reasons To Accept:**

This paper proposes a novel approach based on dynamic pruning of hidden dimensions to reduce the model size. Experimental results show improvements in efficiency compared with some compressed models.

**Reasons To Reject:**

A. The paper is motivated by the expectation that dynamic subnetwork selection leads to better efficiency. However, the paper does not compare the proposed model with static compression methods comprehensively, such as structured pruning [1], leaving the motivation unconvincing.

B. Moreover, existing baselines for comparison need to be updated including [2].

C. The reasons of design choices, such as hardness, adaptation methods, and pooler-unpooler structures, are unclear due to the lack of theoretical or empirical evidences.

[1] Xia et al., Structured Pruning Learns Compact and Accurate Models, ACL 2022.

[2] Zhang et al., PCEE-BERT: Accelerating BERT Inference via Patient and Confident Early Exiting, NAACL Finding 2022.

**Reproducibility:**

3: Could reproduce the results with some difficulty. The settings of parameters are underspecified or subjectively determined; the training/evaluation data are not widely available.

**Reviewer Confidence:**

4: Quite sure. I tried to check the important points carefully. It's unlikely, though conceivable, that I missed something that should affect my ratings.

---

> ### Author Rebuttal · Authors · 2023-08-29
>
> We would like to thank the reviewer for their time and valuable feedback. Please read our comment on positioning our paper in the literature which is available [here](https://openreview.net/forum?id=M1GRz46Ahz&noteId=GlysqKUbZV).
>
> We address the concerns and questions of the reviewer as follows:
>
> **A. The paper is motivated by the expectation that dynamic subnetwork selection leads to better efficiency. However, the paper does not compare the proposed model with static compression methods comprehensively, such as structured pruning, leaving the motivation unconvincing.**
>
> We acknowledge that structured pruning is an effective method for improving inference efficiency of models. Nonetheless, it is *static* and the main drawback of static compression methods is that for each compute constraint a new model should be trained.
> Dynamic approaches, including ours, enable us to attain a broader span of efficiency-accuracy trade-offs. These approaches have the capability to restore full accuracy by directing samples towards the biggest sub-network or to maximize efficiency by routing them to the most efficient sub-network dynamically based on the deployment constraints. Furthermore, in this work as a short paper, we maintained a focused exploration within the particular category of dynamic and sample adaptive inference methods.
>
>
> **B. Moreover, existing baselines for comparison need to be updated including [2].**
>
> We are introducing SHARCS as a new class of adaptive inference methods which uses **width reduction** and a **novel sample hardness-based routing**. To demonstrate its potential, we conduct a comparison between SHARCS and various existing adaptive depth inference methods. It is important to recognize that our adaptive width approach can be used in conjunction with adaptive depth and token reduction methods. Please refer to the rebuttal (first table in the rebuttal available at this [link](https://openreview.net/forum?id=M1GRz46Ahz&noteId=kN68g9XLd3)) where we show that SHARCS can reduce the inference FLOPS of Transkimer by near 40% with the negligible accuracy drop of only ~0.6%.
> Regarding PCEE-BERT, unfortunately, the source code is not available and reproducing their numbers and comparing with them given the short span of rebuttal period does not seem feasible.
>
> **C. The reasons of design choices, such as hardness, adaptation methods, and pooler-unpooler structures, are unclear due to the lack of theoretical or empirical evidences.**
>
> **Hardness**: There are two main approaches in measuring hardness in the literature: Confidence-based hardness score [3] and entropy based hardness score [4]. We did an ablation on deciding between these metrics in appendix E.0.4 (line 710) and figure 8. The figure suggests that confidence based hardness gets better accuracies on MNLI dataset for FLOPs values smaller than 20 TeraFLOPs while on FLOPs values larger than 20 TeraFlops both of the metrics get comparable results.
>
> **Pooler-unpooler**: Pooler can be any module that reduces the dimensionality of the hidden state by the chosen reduction factor. Here are the main pooler modules one can consider:
>
> 1. Taking the first $ r * d $ dimensions of the input where $r$ is the reduction factor and $d$ is model dimension (e.g. if
> $ r=0.25 $
> and $d=768$, we take the first $ 768 * 0.25 = 192 $ dimensions of the hidden state as the input to the adaptive component). This is the approach we take in the paper.
>
> 2. Using some other segment from the input. For example, the last dimensions or any random subset of dimensions in the input. We expect that this approach gets the same accuracy as the first approach. Previous related work in the literature observed that information is uniformly diffused in different segments of the hidden states whether they are drawn from the last, random, or initial dimensions [1, 2]. This is also intuitive as there is no inductive bias in the learning process that make different dimensions encode information differently and the model is not aware of the order of dimensions.
>
> 3. Projecting the input into a vector with a smaller dimensionality using a linear layer. We opt this one out as it adds to the computational complexity which goes against our goal of maintaining computational efficiency within the model.
>
> 4. One other option that we considered while designing the pooler was splitting the hidden state vector into sub-vectors with equal sizes and compute a weighted sum of them. For example, if the reduction factor is $0.25$ and model dimension is $768$, we chunk the hidden states into four $192$-d vectors $v_1$ to $v_4$ and take the sum $ w_1 . v_1 + … + w_4.v_4 $ where $ (w_1, w_2, w_3, w_4) $ is the weight parameter. This will only add four parameters to the method which is much more efficient than using a linear layer. The following table shows the results of comparison between SHARCS that uses first $ r * d$ dimensions and SHARCS that takes a weighted sum of chunks of hidden state. We average the results for threshold values from $0.1$ to $0.9$ for Roberta Base on QNLI while placing router after the third layer
> | Method | QNLI Avg. Acc. |  TeraFLOPs|
> |----------|----------|----------|
> | SHARCS (weighted sum of chunks)    | $82.09$   | $10.41$  |
> | SHARCS (first dimensions)    | $82.45$   | $9.97$  |
> | Roberta base   | $92.8$   | $24.5$ |
>
> The table suggests that SHARCS that uses the initial dimensions outperforms SHARCS that takes the weighted sum of the chunks in terms of accuracy FLOPs trade-off.
>
> Note that the unpooler is the linear layer before the classifier layer. In fact, it is common to have one layer which operates on CLS token of BERT based models before the classifier layer [5]. We reduce the input dimension of this pre-existing layer according to the chosen reduction factor and do not change the output dimension (i.e. the output dimension remains 768 which is the dimensionality of base models in the BERT family).
>
> [1] Kusupati, A., Bhatt, G., Rege, A., Wallingford, M., Sinha, A., Ramanujan, V., Howard-Snyder, W., Chen, K., Kakade, S., Jain, P., & others (2022). Matryoshka Representation Learning. In Advances in Neural Information Processing Systems.
>
> [2] Daniel Soudry, Elad Hoffer, Mor Shpigel Nacson, Suriya Gunasekar, & Nathan Srebro. (2022). The Implicit Bias of Gradient Descent on Separable Data.
>
> [3] Roy Schwartz, Gabriel Stanovsky, Swabha Swayamdipta, Jesse Dodge, & Noah A. Smith. (2020). The Right Tool for the Job: Matching Model and Instance Complexities.
>
> [4] Ji Xin, Raphael Tang, Jaejun Lee, Yaoliang Yu, & Jimmy Lin. (2020). DeeBERT: Dynamic Early Exiting for Accelerating BERT Inference.
>
> [5] Jacob Devlin, Ming-Wei Chang, Kenton Lee, & Kristina Toutanova (2018). BERT: Pre-training of Deep Bidirectional Transformers for Language Understanding. CoRR, abs/1810.04805.

---

### Meta-Review · Area_Chair_ikU3 · 2023-09-18

**Recommendation:** 2

**Metareview:**

The paper presents SHARCS, a method to accelerate Transformer inference using a mechanism that adaptively adjusts the width of Transformer layers based on the perceived difficulty of the input samples.

There was a general concern raised by multiple reviewers that the approach was compared against only a subset of dynamic inference techniques (early-existing approaches) which was considered strange as SHARCS is not an early-exiting approach. The authors provided a general note in the rebuttal that they are focusing on just adaptive approaches (not static methods like pruning). However, the paper would have been better positioned if the initial comparison was against other (more recent) adaptive approaches. In the rebuttal, the authors provide additional results against one token reduction technique (Transkimer).

Another concern raised by the reviewers was a general explanation of why this approach achieves a good trade-off. In the rebuttal, the authors explain that the improvement is due to the introduction of routing based on sample hardness. Additional experiment settings (or ablation studies beyond 4.4) on this main contribution would make the key novelty more clear.

---

### Decision · Program_Chairs · 2023-10-07

**Decision:**

Accept-Findings

**Comment:**

The paper presents SHARCS, a method to accelerate Transformer inference using a mechanism that adaptively adjusts the width of Transformer layers based on the perceived difficulty of the input samples.

There was a general concern raised by multiple reviewers that the approach was compared against only a subset of dynamic inference techniques (early-existing approaches) which was considered strange as SHARCS is not an early-exiting approach. The authors provided a general note in the rebuttal that they are focusing on just adaptive approaches (not static methods like pruning). However, the paper would have been better positioned if the initial comparison was against other (more recent) adaptive approaches. In the rebuttal, the authors provide additional results against one token reduction technique (Transkimer).

Another concern raised by the reviewers was a general explanation of why this approach achieves a good trade-off. In the rebuttal, the authors explain that the improvement is due to the introduction of routing based on sample hardness. Additional experiment settings (or ablation studies beyond 4.4) on this main contribution would make the key novelty more clear.